# Chemical, Physical, and Mechanical Properties and Microstructures of Laser-Sintered Co–25Cr–5Mo–5W (SP2) and W–Free Co–28Cr–6Mo Alloys for Dental Applications

**DOI:** 10.3390/ma12244039

**Published:** 2019-12-04

**Authors:** Yoshimitsu Okazaki, Akira Ishino, Shizuo Higuchi

**Affiliations:** 1National Institute of Advanced Industrial Science and Technology, 1–1 Higashi 1–chome, Tsukuba, Ibaraki 305–8566, Japan; 2IDS Co., Ltd., 3–5–4 Hongo, Bunkyo–ku, Tokyo 113–0033, Japan; a-isino@idscoltd.jp; 3Graduate School of Oral Health Sciences, Osaka Dental University, 1–4–4 Makinohonmachi, Hirakata City, Osaka 573–1144, Japan; shizuo-higuchi@labowada.co.jp

**Keywords:** cobalt–chromium–molybdenum alloys, laser sintering, microstructure, tensile property, fatigue property, physical property, dental prostheses

## Abstract

We examined the chemical, physical, and mechanical properties and microstructures of laser-sintered Co–25Cr–5Mo–5W (SP2) and W–free Co–28Cr–6Mo alloys. The tensile and fatigue properties of the laser-sintered Co–Cr–Mo alloys were extremely superior to those of dental-cast alloys. The ultimate tensile strength (σ_UTS_) and total elongation (T.E.) were close to those of hot-forged Co–28Cr–6Mo alloys. The fatigue strengths (σ_FS_) at 10^7^ cycles of the 90°-, 45°-, and 0°-direction-built Co–28Cr–6Mo alloys were ~500, ~560, and ~600 MPa, respectively. The ratio σ_FS_ /σ_UTS_ was ~0.4. These superior mechanical properties were attributed to the fine π-phase particles in the grains and grain boundaries of the fine face–centered cubic (fcc) matrix formed owing to the rapid solidification. The chemical composition of 20-times-laser-sintered Co–Cr–Mo alloy without the virgin powder added was approximately the same as that of the alloy laser-sintered with the virgin powder. σ_FS_ of the 90°-direction-built alloys after laser sintering 20 times was also ~500 MPa. σ_UTS_ of hot-forged Co–28Cr–6Mo alloys decreased with increasing annealing temperature, whereas T.E. increased. For the Co–Cr–Mo alloys annealed at 1000 to 1150 °C for 30 min after laser sintering, the rates of decrease in σ_UTS_ were small. σ_FS_/σ_UTS_ increased to near those of annealed Co–28Cr–6Mo alloys after hot forging. The durability of clasps fabricated by laser sintering was superior to that of dental-cast clasps.

## 1. Introduction

Dental prostheses require biomechanical compatibility as well as biological safety. Therefore, cobalt–chromium–molybdenum (Co–Cr–Mo) alloys with excellent mechanical properties and structural stability have been widely used for dentures, metal frames, bridges, partial dentures, complete dentures, and implant superstructures in dentistry. Co–Cr–Mo alloys have also been used for total hip and knee replacements in orthopedics, with both cast and wrought materials used. Thus, Co–Cr–Mo alloys are essential for medical applications and have been standardized by the International Organization for Standardization (ISO) [1,2], American Society for Testing and Materials (ASTM) [3,4], and Japanese Industrial Standards (JIS) [5,6,7].

Computer–aided design and computer–aided machining (CAD/CAM) technology has rapidly improved with the remarkable development of digital technology. Dental prostheses with superior mechanical properties and higher fitness accuracy can be fabricated using CAD/CAM as compared with those obtained using the conventional casting method [8]. Metal blocks are mostly used in the milling process using the CAM system to fabricate prosthetic frameworks. However, in the case of using this CAM milling system, it is not easy to fabricate such frameworks with complicated shapes, and long processing times are required [8].

Additive manufacturing (AM), also known as three-dimensional (3D) printing, can be used to fabricate 3D objects in a single stage directly from their CAD. With AM technology, one can form products layer by layer on the basis of sliced data from the 3D design. Laser sintering provides superior dimensional accuracy and material quality of the fabricated parts. In the conventional casting process, materials constantly expand and contract during the formation of wax patterns, investment, and casting. Relatively dense dentures free of blow holes are formed with 3D printing technology, reducing the occurrence of problems such as the breakage of dentures caused by casting defects. In the laser sintering process, a laser is scanned along metal powders on the basis of slice data to obtain a layer of a product. The powder for the next layer is placed on the melted layer, and the laser is again scanned on the basis of the data for the next slice [9,10,11]. Along with the rapid progress of 3D layer manufacturing technologies in the medical field, the development of orthopedic implants customized to the skeletal structure and symptoms of each patient is now possible [12,13,14].

To obtain regulatory approval for dental prostheses and orthopedic devices produced by 3D layer manufacturing in Japan, evaluation of the chemical composition, melting point, microstructure, tensile property, immersion property, and fatigue property of the base metal and the durability of the devices is desired [15,16]. To improve the yield of a powder, it is necessary to examine the effect of the number of repeated uses of a powder in the laser sintering process. IDS Co., Ltd. (Tokyo, Japan), a dental material supplier, acquired Pharmaceutical Affairs Approval for the use of SP2 (Co–25%Cr–5%Mo–5%W) and Co–28%Cr–6%Mo powders (here and hereafter values in alloy compositions indicate mass%) as dental materials for 3D layer manufacturing from the Japanese Ministry of Health, Labor, and Welfare on 27 April 2018 and 10 September 2019 (Approval Nos. 23000BZX00121000 and 30100BZX00137000), respectively [17,18]. Co–25Cr–5Mo–5W alloy satisfies ISO 22674, JIS T 6121, and JIS T 6115 for dental materials [2,5,6]. In particular, Co–Cr–Mo containing tungsten (W) is commonly used for porcelain baking with ceramics. Co–28Cr–6Mo alloys have been used worldwide for dental prostheses and orthopedic devices and standardized by ISO, JIS, and ASTM [1,4,6].

Moreover, in AM technology, heat treatment is also required to remove the thermal strain accumulated during laser sintering. However, there have been few studies on the fatigue properties of prosthesis materials and the effects of heat treatment on these properties.

In this study, we evaluated the chemical composition, physical properties, microstructure, immersion property, tensile property, and fatigue strengths of Co–25Cr–5Mo–5W (SP2) and Co–28Cr–6Mo alloys manufactured using the appropriate powders by laser sintering. The effect of the number of repeated uses of the same powder on these properties was also investigated. In addition, the effects of heat treatment on the tensile property and fatigue strengths of these Co–Cr–Mo alloys as well as their microstructure were examined. Moreover, the fatigue strengths of Co–25Cr–5Mo–5W alloy clasps fabricated by laser sintering were compared with those of dental-cast clasps. The results obtained in this study are expected to be useful for the development of highly biocompatible dental prostheses using 3D layer manufacturing.

## 2. Experimental Procedure

### 2.1. Test Samples

Co–25Cr–5Mo–5W (EOS SP2) and Co–28Cr–6Mo (EOS MP1 and Sanyo S1) alloy powders were prepared by a nitrogen gas atomization process (EOS GmbH, Krailling, Germany and Sanyo Special Steel Co., Ltd., Hyogo, Japan). The chemical compositions of these powders are shown in Table 1. Figure 1 shows (a) the particle size distribution and (b) scanning electron microscopy (SEM) images of the powders. Figure 1a also shows the D_10_, D_50_, and D_90_ particle sizes corresponding to 10%, 50%, and 90% of the cumulative distribution, respectively. The particle size distributions of the three powders show the same tendency. In the actual laser sintering of dental prostheses, a certain percentage of virgin powder is added. The number of repeated uses of the same powder of up to 20 times in this work was decided in consideration of the effect of a small amount of residual powder.

Cylindrical specimens with a diameter of 9 mm and a height of 50 mm were fabricated by laser sintering (EOSINT M 270 and EOSINT M 290) on support materials using the three atomized Co–Cr–Mo powders. To investigate the effect of the building direction of laser sintering, the building direction was set to 0° (hereafter, 0° direction), 45° (45° direction), and 90° (90° direction) for the base plate using the three Co–Cr–Mo powders, as shown in Figure 2a.

The laser beam power (P) and the hatch spacing between scan passes (H) were 120 to 220 W and 0.08 to 0.1 mm, respectively. The laser scan speed (V) and powder stacking (deposited layer) thickness (T) were fixed at values from 600 to 1200 mm/s and 0.02 to 0.04 mm, respectively. The laser spot diameter was 0.1 to 0.5 μm. The volumetric energy density (E) = P/(H·T·V) was 60 to 130 J/mm^3^. Cylindrical specimens built by laser sintering were cut from the support materials. Moreover, the clasps shown in Figure 2b were fabricated by the same laser sintering process and their durability after polishing was evaluated.

AICHROM MB (Co–25Cr–6.5Mo–5W), AICHROM (Co–32Cr–6Mo) (both from IDS Co., Ltd.), WISIL M (Co–28Cr–6Mo) (DeguDent GmbH, Krailling, Germany), and wrought (hot-forged) Co–28Cr–6Mo (Carpenter Technology Cor., Reading, PA, USA.) were used for comparison. The mechanical test specimens of AICHROM MB, AICHROM, and WISIL M shown in Figure 3c were also prepared for conventional dental casting. Moreover, a cemented total hip prosthesis stem (Zimmer Biomet, Tokyo, Japan, VerSys, 00-7653-012-01), which has excellent durability in clinical use, was used for comparison with wrought (hot-forged) Co–28Cr–6Mo.

To investigate the effects of heat treatment on the mechanical properties and microstructure, the laser-sintered SP2 and MP1 Co–Cr–Mo alloys built in the 90°, 45°, and 0° directions on the base plate were heat-treated at 750, 900, 950, 1000, 1050, 1100, and 1150 °C for 0.5 or 1 h followed by water quenching. For comparison, the hot-forged Co–28Cr–6Mo alloys (Carpenter Technology Cor.) were heat-treated under the same conditions.

### 2.2. Evaluation of Physical Properties

The solidus and liquidus temperatures of the three Co–Cr–Mo alloys fabricated by laser sintering were measured by differential thermal analysis (DTA) in accordance with JIS K 0129 [19]. Test specimens of 3 mm diameter and 1.5 mm height were cut from the alloy specimens and the cast materials used for comparison. Heat flows in DTA were measured (Rigaku, DSC8270, Akishima, Japan) at a heating rate of 10 °C/min in Ar at a flow rate of 50 mL/min. The densities (ρ) of powder and laser-sintered Co–Cr–Mo alloys were measured by weighing in a liquid in accordance with JIS Z 8807 [20]. The average linear expansion coefficient was measured using test specimens of 3 mm diameter and 15 mm length at a heating rate of 5 °C/min in Ar at a flow rate of 100 mL/min and calculated from the slope at each temperature in the range from 50 to 500 °C in accordance with JIS Z 2285 [21]. The Vickers hardness (Hv) of each Co–Cr–Mo alloy at room temperature was measured at a load of 1 kg.

### 2.3. Microstructural Observation

The laser-sintered SP2, MP1, and S1 Co–Cr–Mo alloys, and the heat-treated Co–Cr–Mo alloys after laser sintering were embedded in resin and polished to a mirrorlike finish with 200 to 4200 grit waterproof emery paper and an oxide polishing (OP–S) suspension. Then, each specimen was etched with aqua regia. The microstructures of the Co–Cr–Mo alloys were analyzed by optical microscopy (Nikon ECLIPSE LV150, Nikon, Tokyo, Japan) and by transmission electron spectroscopy (TEM, Hitachi H–800 and HF–2000 Hitachi High–Technologies Cor., Tokyo, Japan; acceleration voltage, 200 kV) with energy dispersive X–ray spectroscopy (EDS 2008 ver. 1.2 RevE, Hitachi High–Technologies Cor., Tokyo, Japan). After etching, the surfaces of the Co–Cr–Mo alloys were observed by optical microscopy at magnifications of 50× and 400×. The TEM observations were performed using disc–shaped specimens of 3 mm diameter, which were prepared by electrolytic polishing with 10 vol% perchloric acid+10 vol% methanol+glacial acetic acid solution at 30 V, 70 mA, and 0 °C. The transverse cross–sectional structure was observed by TEM at magnifications of 15,000× and 60,000×. The precipitates were also observed by scanning transmission electron microscopy (STEM, Talos Thermo Fisher Scientific, Tokyo, Japan, F200X; acceleration voltage, 200 kV) with EDS (Super–X). The specimens used in STEM observations were prepared by electrolytic polishing with 10 vol% perchloric acid + 90 vol% ethanol solution. The precipitates were electrolytically extracted in 20 vol% concentrated HCl + 80 vol% ethanol solution at approximately 0 to 0.2 V vs a saturated calomel electrode (SCE) using a speed analyzer (Fujiwara Scientific Co., Ltd., Tokyo, Japan, FV–158). A hydrophobic polytetrafluoroethylene (PTFE)–type membrane filter (Toyo Roshi Kaisha, Ltd., Tokyo, Japan) of 47 mm diameter with 0.1 μm pore size was used to collect the extract. X–ray diffraction (XRD) was then performed with a Co tube (Rigaku, SmartLab, Tokyo, Japan; wavelength, 0.179 nm; tube voltage, 40 kV; tube current, 135 mA) over a scan range (2θ) of 20 to 130°. The Co, Cr, Mo, Si, Mn, N, and C concentrations in the extracted residue were measured in accordance with JIS G 1258–2 [22], JIS G 1228 [23], and JIS G 1211–3 [24]. The fracture surfaces after the tensile and fatigue tests were observed by SEM.

### 2.4. Static Immersion Test

The laser-sintered Co–Cr–Mo alloys were subjected to immersion tests in accordance with ISO 10271 [25]. Plate specimens (*n* = 2), each with dimensions of 15 mm × 32 mm × 1 mm, were laser-sintered and dental-cast for each Co–Cr–Mo alloy. Immersion tests were conducted at 37 °C using 0.1 mol/L lactic acid (LA) + 0.1 mol/L NaCl solution (LA-NaCl solution, Ph = 2.3). The plate specimens were surface-finished with sheets of waterproof emery paper of 240, 600, 800, and 1000 grit under running water, and then ultrasonically cleaned. Two plate specimens of each alloy were separately placed in polypropylene bottles. A 20 mL aliquot of the LA-NaCl solutions (surface area of specimen (cm^2^): liquid volume (mL)=1:1) was then poured into the polypropylene bottles, each containing a plate specimen.

The concentrations of various metals released into LA-NaCl solution were determined in ppb (ng/mL) by inductively coupled plasma mass spectrometry (ICP–MS, PerkinElmer, NexION, Kanagawa, Japan). The Cr, Mo, and Co concentrations were analyzed by ICP–MS. The isotopic mass numbers were selected so as to minimize the influence of the matrix: Cr, 52; Mo, 95; Co, 59; W, 182. The analytical detection limits under these conditions were all below 0.01 μg/mL.

An LA-NaCl solution without a metal specimen was incubated under similar conditions and used for the blank test. The amount of metal released (µg/cm^2^) was estimated using the following formula: (amount of solution: 20 mL) × [(metal concentration in each test solution) – (mean metal concentration in blank test with three bottles)]/(surface area of specimen). The mean amount of each metal released and the standard deviation were calculated for two specimens.

### 2.5. Room-Temperature Tensile Tests

To estimate the mechanical properties of the laser-sintered, forged, and heat-treated specimens and the conventional dental-cast alloys, tensile tests were conducted at room temperature in accordance with JIS Z 2241 [26]. Figure 3 shows the dimensions of each specimen used for the room-temperature tensile and fatigue tests. Figure 3a–c show the uniform rod specimens used for the tensile test. Figure 3a shows the miniature uniform rod specimen used for the tensile test cut from four cylindrical specimens with a diameter of 9 mm and a height of 50 mm. The tensile test specimen (rod diameter, 5 mm; gauge length, 25 mm) shown in Figure 3b was cut from the center of a hot-forged Co–28Cr–6Mo alloy rod with its longitudinal (L) direction parallel to the hot–forging direction. The tensile test specimen shown in Figure 3c was dental-cast using AICHROM MB, AICHROM, or WISIL M.

The tensile test specimens were pulled at a crosshead speed of 0.5% of the gauge length (GL)/min until the proof stress reached 0.2%. The crosshead speed was then changed and maintained at 3 mm/min until the specimen fractured. The 0.2% proof stress (σ_0.2%PS_), ultimate tensile strength (σ_UTS_), total elongation (T.E.), and reduction in area (R.A.) were measured. The mean and standard deviation were calculated from the results of at least four specimens.

### 2.6. Fatigue Tests

To investigate the effects of laser sintering and heat treatment after laser sintering on fatigue strength, fatigue tests were conducted at room temperature in accordance with JIS T 0309 [27]. Figure 3d shows the miniature hourglass-shaped rod specimen used for fatigue tests, which was cut from a cylindrical as-built specimen and from specimens heat-treated under various conditions after laser sintering. The specimens of hot-forged Co–28Cr–6Mo alloy used for the fatigue test, as shown in Figure 3e, were also machined with their longitudinal direction parallel to the hot–rolling direction. Specimens of dental-cast AICHROM MB, AICHROM, and WISIL M Co–Cr–Mo alloys, as shown in Figure 3c, were also used for fatigue tests. To remove the inner strain generated on the surfaces of the specimens during the manufacturing process, the surfaces were fully ground using 600 grit waterproof emery paper in the direction parallel to the test specimen.

The fatigue tests were carried out using an electrohydraulic servo testing machine with a sine wave at a stress ratio R [minimum cyclic stress (σ_min_)/(maximum cyclic stress (σ_max_)] of 0.1 and a frequency of 10 Hz in air. To obtain profiles of the relationship between σ_max_ and the number of cycles to failure N (S–N curves), the specimens were cycled at various constant maximum cyclic loads up to N = 10^7^ cycles, at which the specimens remained intact. The fatigue strengths at 10^7^ cycles (fatigue limit, σ_FS_) were measured from the S–N curves. That is, σ_FS_ was determined from the S–N curve as the maximum stress at which a specimen does not break after cyclic loading for 10^7^ cycles (see Figure 8).

The fatigue strengths of SP2 Co–Cr–Mo alloy clasps, as shown in Figure 2b, fabricated by laser sintering were compared with those of dental-cast AICHROM MB Co–Cr–Mo alloy clasps. The fatigue tests of clasps were conducted with a jig, as shown in Figure 2c, under sinusoidal loading at a stress ratio R of 0.1 and a frequency of 3 Hz in air. To obtain the relationship between the maximum cyclic load (L) and the number of cycles to failure N (L–N curves), the specimens were cycled at various constant maximum cyclic loads up to N = 10^6^ cycles.

## 3. Results and Discussion

### 3.1. Chemical Compositions and Physical Properties

Table 1 shows the chemical compositions of the laser-sintered SP2, MP1, and S1 Co–Cr–Mo alloys with those of the virgin powders (hereafter, once-sintered Co–Cr–Mo), and the compositions after laser sintering 20 times (20-times-sintered Co–Cr–Mo) with the same Co–Cr–Mo powders without the virgin powder added. The changes in the chemical composition of 20-times-sintered Co–Cr–Mo alloys were negligible. In particular, almost no increase in the oxygen concentration was observed after sintering 20 times. The concentrations of the harmful elements Be, Cd, and Pd were ≤0.0001, ≤0.0001, and ≤0.001%, respectively.

The densities of once-sintered Co–Cr–Mo alloys with SP2, MP1, and S1 powders were 8.8, 8.4, and 8.4 g/cm^3^, respectively. These values were close to those (8.5 for AICHROM MB, 8.2 for AICHROM, and 8.4 g/cm^3^ for WISIL M) of the conventional dental-cast Co–Cr–Mo alloys. The solidus and liquidus temperatures of these SP2, MP1, and S1 powders measured by DTA were 1362 and 1458, 1351 and 1461, and 1357 and 1463 °C, respectively. The solidus and liquidus temperatures of AICHROM MB, AICHROM, and WISIL M were 1335 and 1410, 1325 and 1370, and 1357 and 1463 °C, respectively. The average linear expansion coefficient of Co–25Cr–6.5Mo–5W alloy manufactured with SP2 was 14.4×10^–6^ K^–1^. The addition of W increases thermal expansion coefficient.

### 3.2. Static Immersion Property of Laser-Sintered Co–Cr–Mo Alloys

The total amounts of metal ions released from the 20-times-sintered SP2 and once-sintered MP1 Co–Cr–Mo alloys were 0.26 ± 0.05 and 0.25 ± 0.05 μg/cm^2^/week, respectively, which are markedly smaller than the maximum amount of 200 μg/cm^2^/week specified in ISO 22674 [2], JIS T 6121 [5], and JIS T 6115 [6], indicating their excellent corrosion resistance.

### 3.3. Microstructure of Laser-Sintered Co–Cr–Mo Alloys

Figure 4 shows optical micrographs of transverse (T) and longitudinal (L) sections of laser-sintered SP2 Co–Cr–Mo alloys (90°, 0°, and 45° directions). For comparison, optical micrographs of dental-cast AICHROM MB and hot-forged Co–28Cr–6Mo alloys are respectively shown in Figure 4g,h. The laser-sintered Co–Cr–Mo alloys had a finer structure than the conventional dental-cast AICHROM MB alloy because of the effects of melting and rapid solidification. The 90°-, 0°-, and 45°-direction-built specimens had similarly fine structures. Similar fine structures were observed in the transverse and longitudinal sections of all other laser-sintered Co–Cr–Mo alloys.

Figure 5 shows TEM images of transverse sections of 90°-direction-built laser-sintered SP2 and MP1 Co–Cr–Mo alloys and dental-cast AICHROM MB. In the dental-cast AICHROM MB, a face–centered cubic (fcc) structure containing slightly dislocated lines was observed. On the other hand, for the laser-sintered SP2 and MP1 Co–Cr–Mo alloys, a fine cell–like structure was formed because of repeated rapid solidification. Many stacking faults formed owing to the applied thermal stress were observed. The precipitates formed in the grains and grain boundaries of the fcc (lattice parameters a = b = c = 0.356 nm) matrix (Figure 5d,e). The results of EDS analysis of a lathlike precipitate are shown in Figure 5g,h. The Cr, Mo, Co, and Si concentrations in the precipitate (π-phase) shown in Figure 5f were 39.4, 14.2, 43.2, and 3.2 at%, respectively. These values were substantially the same as the literature values for the π-phase precipitated in Co–28Cr–6Mo alloys [28]. However, in the electron beam diffraction pattern of the precipitate shown in Figure 5f, the diffraction spots overlap and it cannot be concluded whether the precipitate was of the π-phase. Amorphous particulate Si–Mn–based oxides were present in or near the π-phases.

Figure 6 shows a STEM image, elemental mappings obtained using EDS, and an electron beam diffraction pattern of the plate-shaped precipitate at the grain boundary of 0°–direction laser-sintered MP1 Co–28Cr–6Mo alloy. The precipitate was the π-phase (β Mn structure, a = b = c = 0.633 to 0.640 nm, JCPDS card no. 00–026–0428) as shown in Figure 6g. The Cr and Mo concentrations in the π-phase were higher than those in the matrix. In STEM, analysis accuracy is improved because many electron beam diffraction spots can be obtained compared with TEM. XRD analysis of the precipitate electrolytically extracted from laser-sintered MP1 and SP2 Co–Cr–Mo alloys also revealed that the precipitate was the π-phase (a = b = c = 0.633 nm), as shown in Figure 6h. However, the peak corresponding to the π-phase shown in Figure 6h was not sharp (i.e., broad) because the precipitate was composed of fine particles. According to the electron beam diffraction analysis of the precipitate, the π-phase was also precipitated in laser-sintered SP2 Co–Cr–Mo alloy. The Cr, Mo, Co, Si, Mn, N, and C concentrations in the π-phase obtained by chemical analysis of the precipitate extracted from MP1 were 34.11, 11.54, 31.86, 0.29, 2.87, 1.20, and 18.15 at%, respectively. Small crystal grains were observed in the wrought (hot-forged) Co–28Cr–6Mo alloy, as shown in the TEM image [Figure 5b]. The fine structure in the laser-sintered Co–Cr–Mo alloy was similar to that in the wrought Co–28Cr–6Mo alloy used for cemented total hip prosthesis stems in clinical use.

### 3.4. Mechanical Properties of Laser-Sintered Co–Cr–Mo Alloys

Table 2 summarizes the tensile properties (mean ± standard deviation) of the various laser-sintered Co–Cr–Mo alloys fabricated from SP2, MP1, and S1 powders, and the dental-cast and hot-forged Co–Cr–Mo alloys. Moreover, the effects of the heat treatment on the tensile properties of the laser-sintered and hot-forged Co–Cr–Mo alloys are compared in Table 3.The tensile properties of the laser-sintered Co–Cr–Mo alloys were extremely superior to those of the dental-cast Co–Cr–Mo alloys, which may be due to the reduced number of blow holes and the difference in the microstructure. However, they were inferior to those of the hot-forged Co–28Cr–6Mo alloy. σ_0.2%PS_ of all laser-sintered Co–Cr–Mo alloys satisfied the ≥500 MPa condition specified in ISO 22674 [2]. σ_0.2%PS_ and σ_UTS_ of the 0°-direction-built specimens were higher than those of the 90°-direction-built specimens, whereas T.E. of the 90°-direction-built specimens was larger than that of the 0°-direction-built specimens. The strengths of the 20-times-sintered Co–Cr–Mo alloys were similar to those of the once-sintered Co–Cr–Mo alloys. The tendencies observed in SP2 were also observed in the MP1 and S1 Co–Cr–Mo alloys, as shown in Table 2. The Vickers hardnesses of 20-times-sintered SP2, MP1, and S1 and dental-cast AICHROM MB were 351, 386, 382, and 350, respectively.

Figure 7 shows SEM images of the fracture surfaces of laser-sintered MP1 Co–Cr–Mo alloys after the tensile and fatigue tests. Dimples were observed on all fracture surfaces after the tensile test. A magnification of the rectangular area in Figure 7a is shown in Figure 7b. Similar fracture surfaces were observed on the other laser-sintered Co–Cr–Mo alloys.

### 3.5. Fatigue Strengths of Laser-Sintered Co–Cr–Mo Alloys

Figure 8 shows S–N curves of laser-sintered SP2 Co–Cr–Mo alloys. Figure 9a,b also show S–N curves of the laser-sintered MP1 and S1 Co–Cr–Mo alloys, respectively. As shown in Figure 8, SP2 Co–Cr–Mo alloys were built in the 0°, 45°, and 90° directions to examine the effects of the laser sintering direction on the fatigue strength. To examine the effects of the repeated use of powder on the fatigue strength, 90°-direction-built Co–Cr–Mo alloys were also laser-sintered using the 20-times-sintered SP2 Co–Cr–Mo alloy powder. For comparison, S–N curves of dental-cast AICHROME MB, AICHROME, and WISIL M are shown in Figure 8 and Figure 9a, which show that the fatigue strengths (σ_FS_) of these conventional dental-cast materials at 10^7^ cycles were low (160, 200, and 220 MPa, respectively) because of the presence of blow holes. On the other hand, the fatigue strengths of the 90°-, 45°-, and 0°-direction-built SP2 Co–Cr–Mo alloys were ~500, ~560, and ~600 MPa, respectively, which are sufficiently high. The effects of the laser sintering direction on σ_FS_ of SP2 Co–Cr–Mo alloys were considered to be small because the anisotropy factor of the fatigue strength was 0.83 [(fatigue strength in 90° direction)/(fatigue strength in 0° direction)]. The fatigue strength of the 90°-direction-built specimens of the 20-times-sintered SP2 Co–Cr–Mo alloy was ~500 MPa. As shown in Figure 9a, the fatigue strengths of the 90°-, 45°-, and 0°-direction-built MP1 Co–Cr–Mo alloys were ~500 MPa. The same tendency was also observed for laser-sintered S1 Co–Cr–Mo alloy, as shown in Figure 9b. The fatigue strength of the 90°-direction-built specimens of the 20-times-sintered MP1 Co–Cr–Mo alloy was also ~500 MPa. We would like to perform a statistical analysis of the fatigue strength in the future. Figure 7c,d show SEM images of the fracture surface of a laser-sintered MP1 Co–Cr–Mo alloy after the fatigue test. A fatigue crack developed from the periphery of the sample.

### 3.6. Effects of Annealing on Mechanical Properties and Microstructure of Laser-Sintered Co–Cr–Mo Alloys

Figure 10 shows the effects of heat treatment (annealing at each temperature for 30 min) on the tensile properties obtained using the 90°-direction-built laser-sintered SP2, MP1, and hot-forged Co–28Cr–6Mo alloys. σ_0.2%PS_ and σ_UTS_ of the hot-forged Co–28Cr–6Mo alloy decreased with increasing annealing temperature, whereas T.E. of the hot-forged Co–28Cr–6Mo alloy increased with increasing annealing temperature. The effects of the annealing temperature on the tensile properties of the hot-forged Co–28Cr–6Mo alloy were in agreement with the results reported in the literature [29]. On the other hand, it was found that, in the laser-sintered MP1 Co–Cr–Mo alloys, the decreases in strength were small. σ_0.2%PS_ and σ_UTS_ of both once– and 20-times-laser-sintered SP2 and MP1 Co–Cr–Mo alloys built in the 0° and 45° directions were slightly high, and their T.E. was slightly low. The same tendencies were observed for SP2 and MP1 Co–Cr–Mo alloys heat-treated for 60 min at 1000 and 1150 °C after laser sintering in the 90° direction both once and 20 times. Figure 11 shows the change in the optical microstructure after annealing the laser-sintered SP2 and MP1 Co–Cr–Mo alloys. The structure became coarse with increasing annealing temperature. The optical micrograph of laser-sintered MP1 Co–Cr–Mo alloy after heat treatment at 1150 °C for 30 min was similar to that of the hot-forged Co–28Cr–6Mo alloy shown in Figure 4h. Figure 11e shows TEM images of transverse sections of the laser-sintered MP1 Co–Cr–Mo alloy annealed at 1150 °C for 30 min. Stacking faults that formed owing to thermal stress disappeared after annealing at 1150 °C for 30 min. A few fine precipitates were observed at the grain boundaries. The results of EDS analysis of a precipitate are shown in Figure 11f. The concentrations of O, Al, Cr, and Mn were high in the precipitate. Electron beam diffraction analysis revealed the precipitate and matrix to be (Cr, Mn)_2_AlO_4_ (a = b = c = 0.837 nm) and fcc (a = b = c = 0.355 nm), respectively.

Figure 12 shows the effect of the heat treatment on the fatigue strength of the 90°-direction-built laser-sintered SP2 and hot-forged Co–28Cr–6Mo alloys. σ_FS_ of the hot-forged Co–28Cr–6Mo alloy tended to decrease with increasing heat treatment temperature. The same tendency was observed in the laser-sintered MP1 Co–Cr–Mo alloy. It was found that the σ_FS_ values of 90°-direction-built laser-sintered SP2 and MP1 decreased at a small rate with increasing temperature.

σ_FS_ and the ratios of the fatigue strength at 10^7^ cycles to the ultimate tensile strength (σ_FS_/σ_UTS_) obtained in this work are also summarized in Table 2 and Table 3. The obtained σ_FS_/σ_UTS_ of the laser-sintered specimens was ~0.4, which can be used to estimate the fatigue strength from the tensile strength. It was thus demonstrated that a relatively high fatigue strength can be achieved by laser sintering, which produces a microstructure consisting of a fine fcc phase precipitated in a fine π-phase. σ_FS_/σ_UTS_ of the laser-sintered MP1 Co–Cr–Mo alloy after annealing increased up to 0.5, as shown in Table 3. In particular, it was clear that σ_FS_ of the laser-sintered MP1 Co–Cr–Mo alloy annealed at 1150 °C for 30 min is close to that of wrought Co–28Cr–6Mo alloy annealed at the same temperature.

### 3.7. Durability of Laser-Sintered Co–Cr–Mo Alloy Clasps

The durability of clasps, which have a high risk of breakage because they are subjected to a large amount of bending in the oral cavity, is important in the development of dental prostheses using laser–sintering technology. When a tip of the clasp is placed against an undercut, the clasp covers and supports abutment teeth. Clasps were fabricated by laser sintering with SP2 and by dental-casting with AICHROM MB. Figure 13 shows the L–N curves of laser-sintered SP2 and dental-cast Co–Cr–Mo alloy clasps. The durability of the clasps fabricated by laser sintering was superior to that of conventional dental-cast clasps. From the results obtained in this work, additive manufacturing may be a promising new technology to replace dental casting.

## 4. Conclusions

We evaluated the chemical composition, physical properties, immersion property, microstructure, tensile property, and fatigue strength of laser-sintered Co–25Cr–5Mo–5W (SP2) and Co–28Cr–6Mo alloys for dental applications. In addition, the effects of heat treatment on the tensile property and fatigue strength of these Co–Cr–Mo alloys as well as their microstructure were examined. Moreover, the fatigue properties of Co–25Cr–5Mo–5W alloy clasps fabricated by laser sintering were compared with those of dental-cast clasps.

The changes in the chemical composition of Co–Cr–Mo alloys after laser sintering up to 20 times were negligible. The density and melting point of the laser-sintered Co–Cr–Mo alloys were close to those of the conventional dental-cast alloys. The rates of metal ion release were low, with SP2 and MP1 showing release rates of 0.26 ± 0.05 and 0.25 ± 0.05 μg/cm^2^/week, respectively.

The laser-sintered Co–Cr–Mo alloys had a finer structure than the conventional dental-cast alloys because of the effects of melting and rapid solidification. Precipitates were found in the grains and grain boundaries of the fcc matrix. The precipitates were the π-phase (a = b = c = 0.633 to 0.640 nm). This fine structure in the laser-sintered alloys was similar to that in the hot-forged Co–28Cr–6Mo alloy. It was found that the strength and ductility of laser-sintered Co–Cr–Mo alloys are excellent owing to the dispersion effect of the fine π-phase particles.

The tensile and fatigue properties of the laser-sintered Co–Cr–Mo alloys were extremely superior to those of the dental-cast alloys. The tensile properties were close to those of the hot-forged alloys, although the fatigue strength was lower than that of the hot-forged alloys. The fatigue strengths of the 90°-, 45°-, and 0°-direction-built Co–28Cr–6Mo alloys were ~500, ~560, and ~600 MPa, respectively. The fatigue strength of the 90°-direction-built alloys after laser sintering 20 times was also ~500 MPa. The ratio of the fatigue strength at 10^7^ cycles to the ultimate tensile strength (σ_FS_/σ_UTS_) was ~0.4.

σ_0.2%PS_ and σ_UTS_ of the hot-forged Co–28Cr–6Mo alloys decreased with increasing heat treatment temperature, whereas T.E. increased. On the other hand, for Co–Cr–Mo alloys annealed at 1000 to 1150 °C for 30 min after laser sintering, the rates of decrease in σ_UTS_ were small. The σ_FS_/σ_UTS_ increased to near those of annealed Co–28Cr–6Mo alloys after hot forging.

The durability of clasps fabricated by laser sintering was superior to that of dental-cast clasps. Additive manufacturing may be a promising new technology to replace dental casting.

## Figures and Tables

**Figure 1 materials-12-04039-f001:**
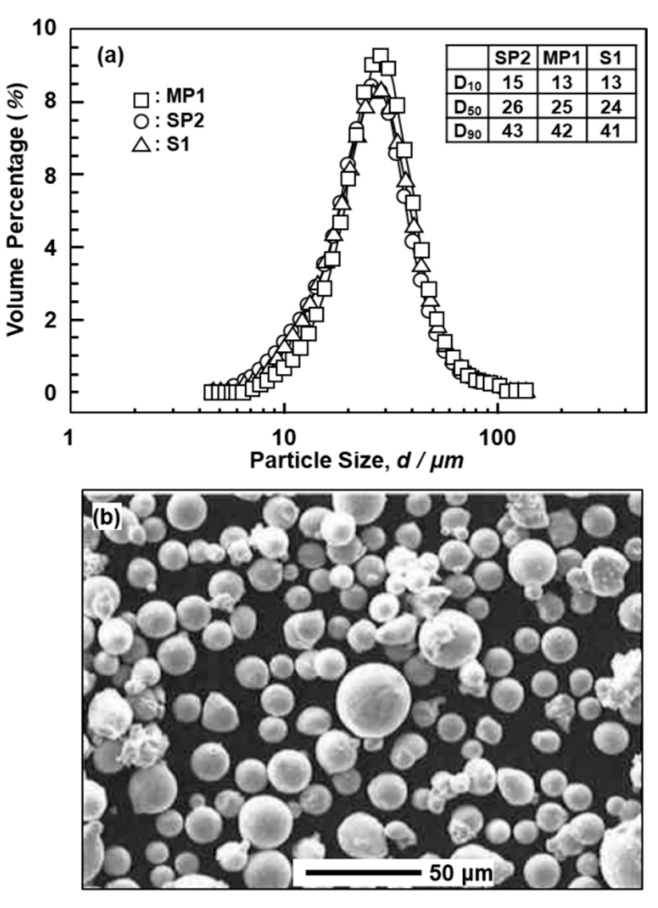
(**a**) particle size distributions of SP2, MP1, and S1 virgin powders; (**b**) micrograph of SP2 powder.

**Figure 2 materials-12-04039-f002:**
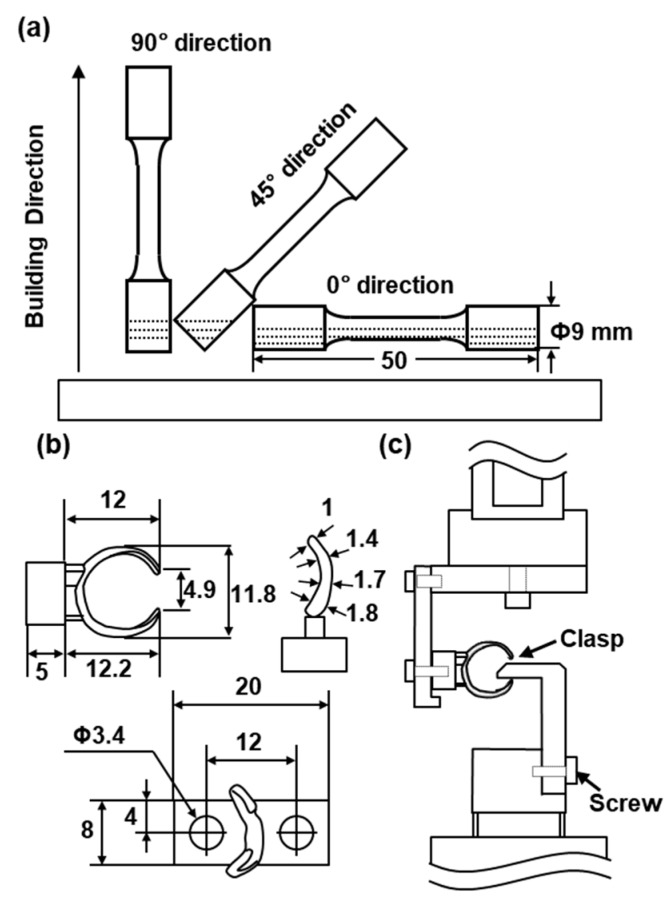
(**a**) building directions of cylindrical specimens; (**b**) dimensions of clasps fabricated by laser sintering and dental casting; (**c**) jig for determining durability of clasp.

**Figure 3 materials-12-04039-f003:**
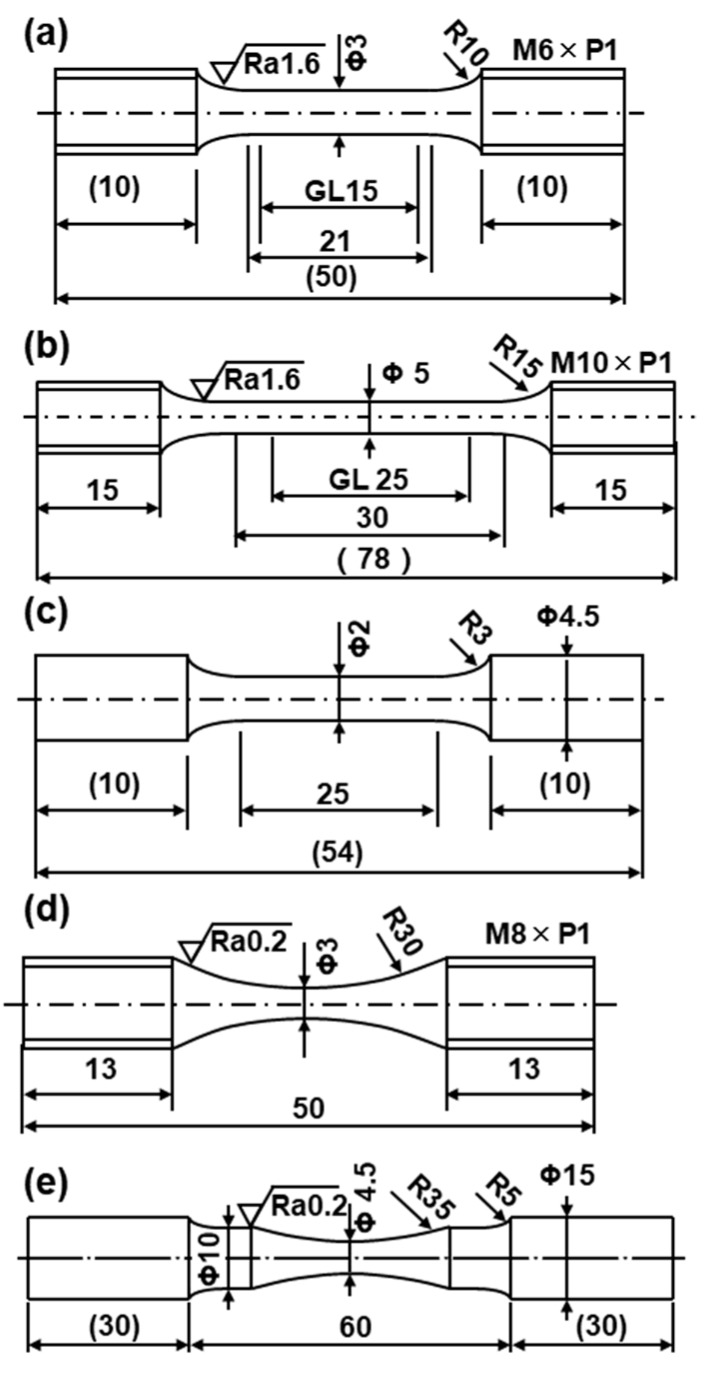
Dimensions of specimens used for room-temperature tensile and fatigue tests: (**a**) miniature test specimen used in a tensile test; (**b**) specimen used in a conventional tensile test; (**c**) dental-cast specimen; (**d**) miniature hourglass–shaped rod specimen used in a fatigue test; (**e**) hourglass–shaped rod specimen used in a conventional fatigue test.

**Figure 4 materials-12-04039-f004:**
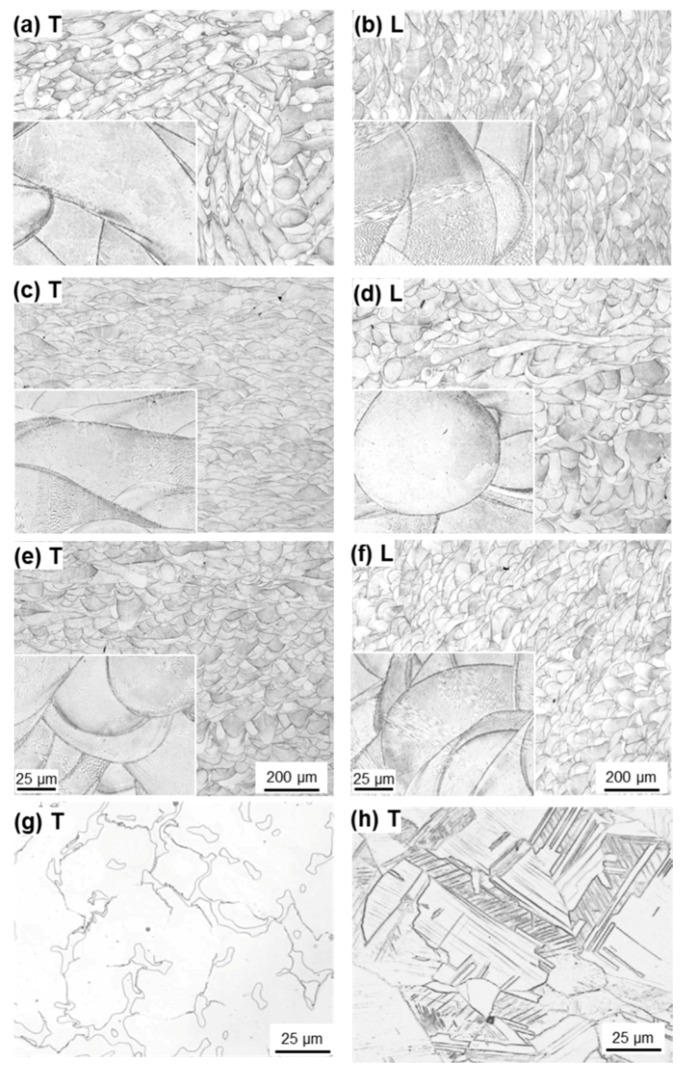
Optical micrographs of laser-sintered SP2 Co–Cr–Mo alloys built in (**a**,**b**) 90°, (**c**,**d**) 0°, and (**e**,**f**) 45° directions; (**a**,**c**,**e**) transverse (T) sections to the building direction and (**b**,**d**,**f**) longitudinal (L) sections perpendicular to the building direction; (**g**) dental-cast AICHROM MB Co–Cr–Mo alloy; (**h**) optical micrograph of cemented hip stem (wrought) Co–Cr–Mo alloy.

**Figure 5 materials-12-04039-f005:**
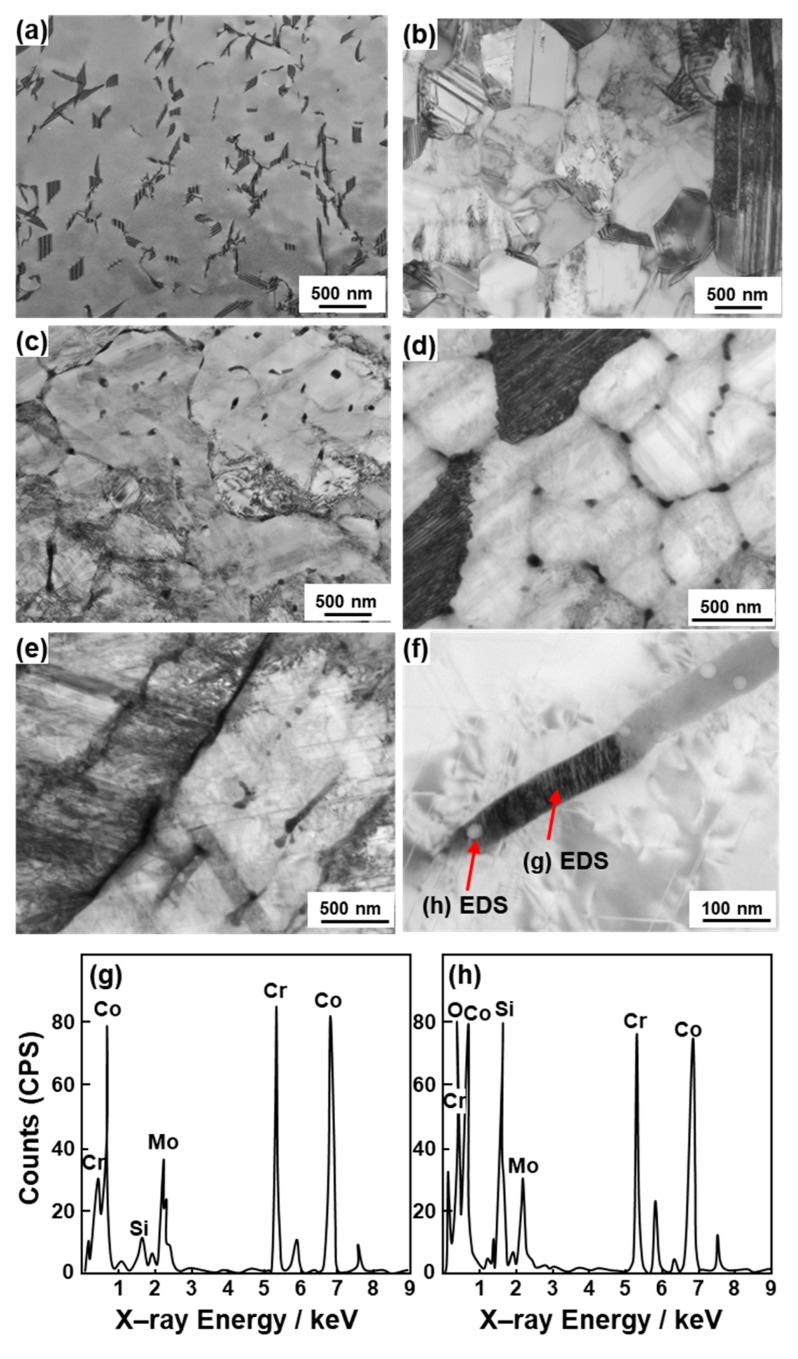
TEM images of transverse sections of (**a**) dental-cast AICHROM MB; (**b**) cemented hip stem and (**c**) laser-sintered SP2; and (**d**,**e**,**f**) laser-sintered MP1 Co–Cr–Mo alloys; (**g**,**h**) energy dispersive X–ray spectroscopy (EDS) patterns of lath precipitate in grain of MP1.

**Figure 6 materials-12-04039-f006:**
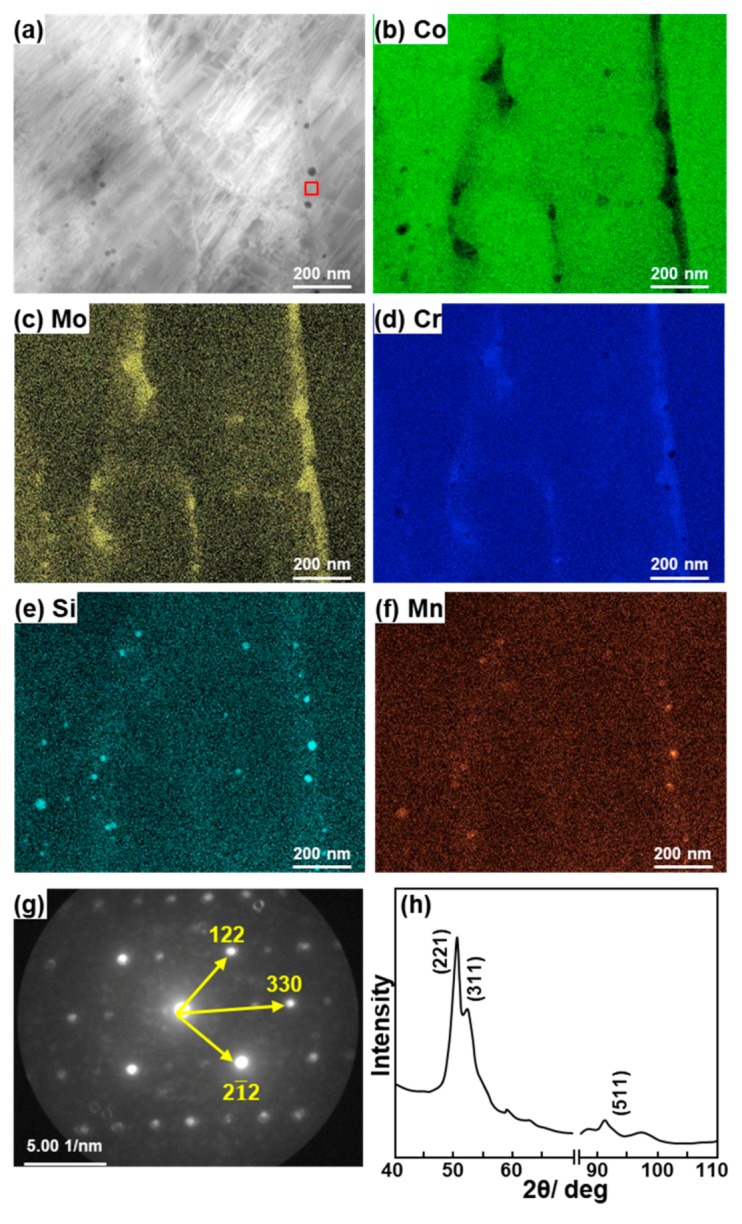
(**a**) Scanning transmission electron microscopy (STEM) image and (**b**–**f**) elemental mappings obtained using EDS for transverse section of 0°-direction-built specimen; (**g**) electron beam diffraction pattern obtained at the location shown by the square in (**a**); (**h**) X–ray diffraction (XRD) pattern of precipitates (π-phase) electrolytically extracted from laser-sintered MP1 Co–28Cr–6Mo alloy.

**Figure 7 materials-12-04039-f007:**
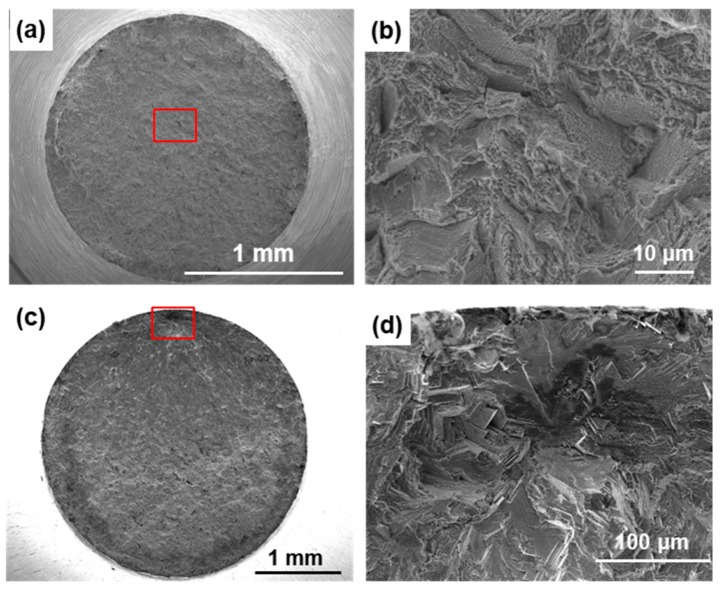
(**a**,**c**) SEM images of fracture surfaces of the tensile–tested and fatigue–tested MP1 Co–Cr–Mo alloys; (**b**) magnification of rectangular area in (**a**), and (**d**) magnification of rectangular area in (**c**).

**Figure 8 materials-12-04039-f008:**
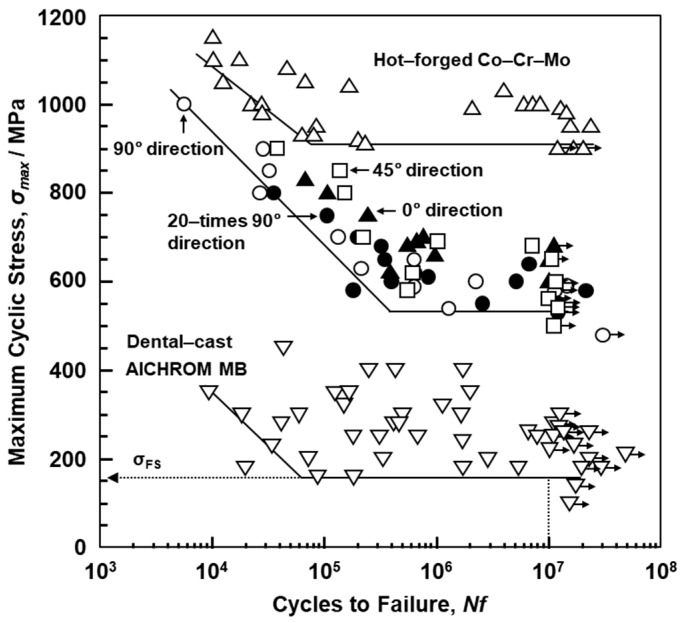
S–N curves of laser-sintered SP2 Co–Cr–Mo alloys, dental-cast AICHROM MB, and hot-forged Co–Cr–Mo alloys.

**Figure 9 materials-12-04039-f009:**
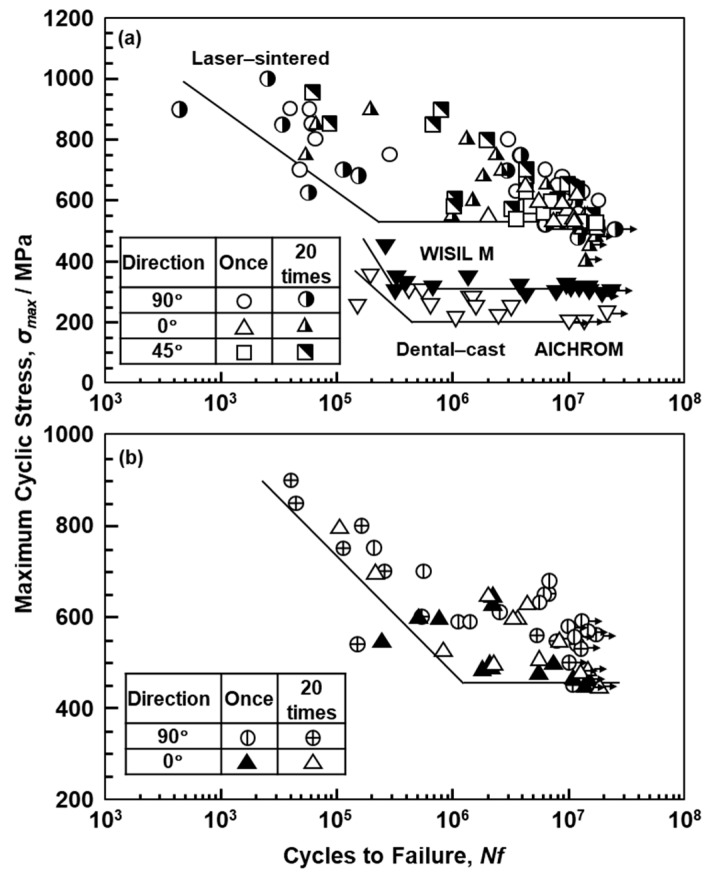
(**a**) S–N curves of laser-sintered MP1 Co–Cr–Mo alloys, dental-cast WISIL M, and AICHROM Co–Cr–Mo alloys; (**b**) S–N curves of S1 Co–Cr–Mo alloys laser-sintered once and 20 times in 90**°** and 0**°** directions.

**Figure 10 materials-12-04039-f010:**
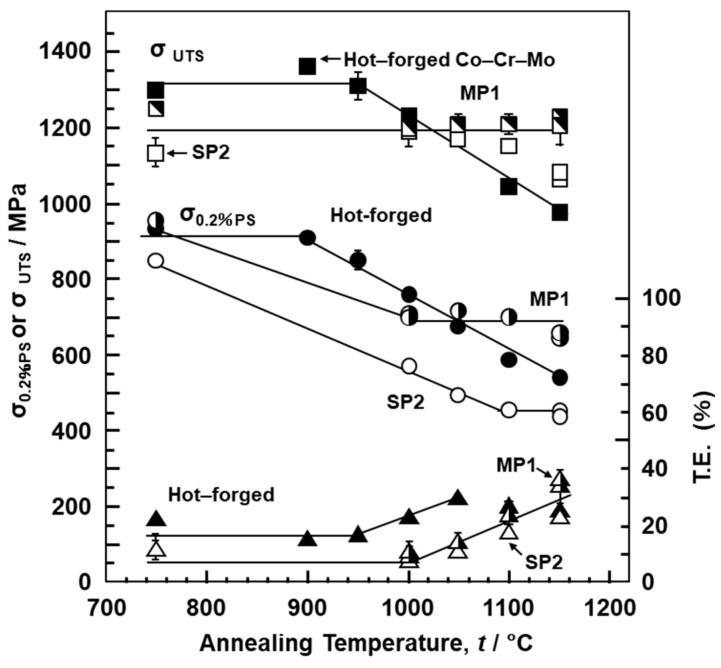
Effects of heat treatment on mechanical properties (σ_0.2%PS_, σ _UTS_, and T.E.) of laser-sintered MP1, SP2, and hot-forged Co–Cr–Mo alloys.

**Figure 11 materials-12-04039-f011:**
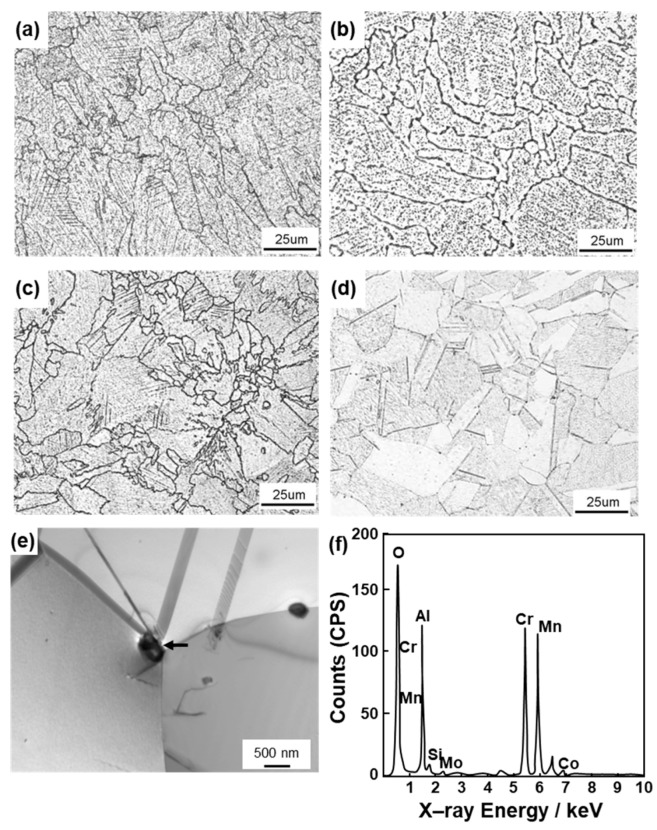
Optical micrographs of (**a**,**b**) SP2 and (**c**,**d**) MP1 Co–Cr–Mo alloys annealed at 1000 and 1150 °C for 30 min after laser sintering. Annealing at 1000 °C (**a**,**c**) and 1150 °C (**b**,**d**). (**e**) TEM image of transverse section of MP1 Co–Cr–Mo alloy annealed at 1150 °C for 30 min. (**f**) EDS pattern of precipitate indicated by the arrow in (**e**).

**Figure 12 materials-12-04039-f012:**
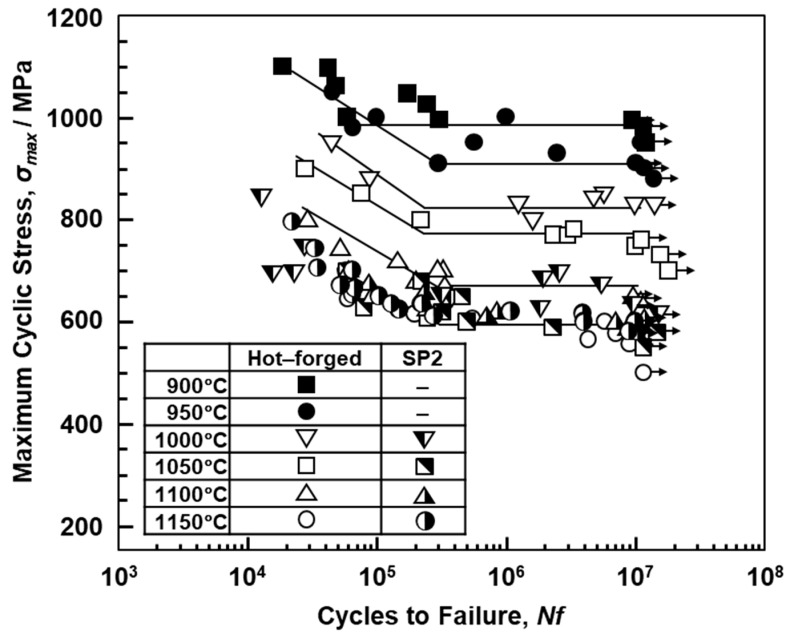
Effect of heat treatments from 900 to 1150 °C for 30 or 60 min on L–N curves of hot-forged and 20-times-laser-sintered SP2 Co–Cr–Mo alloys.

**Figure 13 materials-12-04039-f013:**
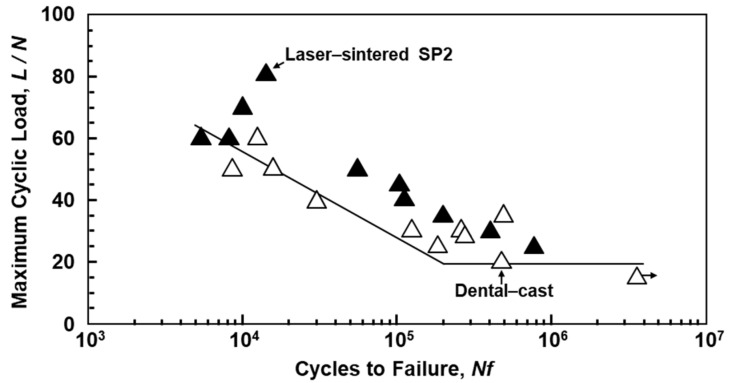
L–N curves obtained for durability of clasps fabricated by laser sintering with SP2 powder and by dental casting with the AICHROM MB Co–Cr–Mo alloy.

**Table 1 materials-12-04039-t001:** Chemical compositions (mass%) of SP2, MP1, and S1 virgin powders and laser-sintered Co–Cr–Mo alloys.

Alloy	Cr	Mo	Ni	Fe	C	N	Mn	Si	W	Co
**SP2 virgin powder**	24.6	5.0	0.021	0.02	0.006	<0.01	<0.001	1.09	5.67	Bal.
**Once-sintered**	24.0	5.16	0.034	0.034	0.008	0.012	0.011	1.05	5.48	Bal.
**20-times-sintered**	24.4	5.05	0.023	0.029	0.007	0.01	<0.001	1.10	5.62	Bal.
**MP1 virgin powder**	27.4	5.88	0.016	0.037	0.054	0.16	0.66	0.81	−	Bal.
**Once-sintered**	27.4	6.03	0.017	0.073	0.13	0.13	0.62	0.81	−	Bal.
**20-times-sintered**	27.5	6.06	0.017	0.039	0.13	0.13	0.61	0.82	−	Bal.
**S1 virgin powder**	27.6	5.85	0.005	0.16	0.086	0.16	0.69	0.67	−	Bal.
**Once-sintered**	27.2	6.27	0.005	0.13	0.095	0.14	0.63	0.67	−	Bal.
**20-times-sintered**	27.4	6.04	0.005	0.11	0.094	0.15	0.54	0.69	−	Bal.

**Table 2 materials-12-04039-t002:** Tensile properties (σ_0.2%PS_, σ_UTS_, T.E., and R.A.), fatigue strength at 10^7^ cycles (σ_FS_), and fatigue ratios (σ_FS_/σ_UTS_) of Co–Cr–Mo alloys manufactured under various conditions.

Specimen	σ_0.2%PS_/MPa	σ_UTS_/MPa	T.E. (%)	R.A. (%)	σ_FS_/MPa	σ_FS_/σ_UTS_
**Laser-sintered**						
**SP2 Once-sintered 90°**	520 ± 5	1164 ± 6	25 ± 1	21 ± 1	500	0.43
**SP2 Once-sintered 0°**	793 ± 6	1310 ± 13	13 ± 2	13 ± 2	600	0.43
**SP2 Once-sintered 45°**	805 ± 5	1309 ± 3	15 ± 1	15 ± 2	560	0.46
**SP2 20-times-sintered 90°**	508 ± 10	1159 ± 11	27 ± 2	23 ± 2	500	0.43
**MP1 Once-sintered 90°**	660 ± 60	1301 ± 16	22 ± 2	21 ± 1	500	0.38
**MP1 Once-sintered 0°**	1029 ± 8	1415 ± 7	13 ± 1	13 ± 2	500	0.35
**MP1 Once-sintered 45°**	948 ± 77	1388 ± 6	14 ± 1	14 ± 2	500	0.36
**MP1 20-times-sintered 90°**	758 ± 9	1303 ± 12	23 ± 2	20 ± 2	500	0.38
**MP1 20-times-sintered 0°**	1034 ± 5	1399 ± 3	12 ± 1	13 ± 3	500	0.36
**MP1 20-times-sintered 45°**	1019 ± 15	1381 ± 10	13 ± 2	14 ± 1	500	0.36
**S1 Once-sintered 90°**	766 ± 4	1269 ± 5	22 ± 1	18 ± 1	500	0.39
**S1 Once-sintered 0°**	1011 ± 8	1391 ± 3	13 ± 1	13 ± 2	500	0.36
**S1 20-times-sintered 90°**	755 ± 10	1265 ± 7	23 ± 2	19 ± 1	500	0.40
**S1 20-times-sintered 0°**	1009 ± 3	1399 ± 5	13 ± 1	12 ± 1	500	0.36
**Dental–cast**						
**AICHROM MB**	580 ± 13	658 ± 21	2 ± 2	6 ± 3	160	0.33
**AICHROM**	512 ± 75	628 ± 60	2 ± 1	5 ± 5	200	0.32
**WISIL M**	722 ± 20	781 ± 133	2 ± 1	2 ± 1	300	0.38
**Hot–forged**						
**Hot–forged Co–Cr–Mo**	903 ± 5	1269 ± 6	24 ± 1	23 ± 1	900	0.71

**Table 3 materials-12-04039-t003:** Effect of annealing temperature on tensile properties (σ_0.2%PS_, σ_UTS_, T.E., and R.A.) and fatigue ratios (σ_FS_/σ_UTS_).

Specimen	σ_0.2%PS_/MPa	σ_UTS_/MPa	T. E. (%)	R. A. (%)	σ_FS_/MPa	σ_FS_/σ_UTS_
**20-times-sintered 90°**						
**SP2 Annealed 750 °C for 1 h**	849 ± 17	1132 ± 37	13 ± 5	14 ± 3	500	0.44
**SP2 Annealed 1000 °C for 0.5 h**	569 ± 14	1188 ± 10	7 ± 1	8 ± 1	620	0.52
**SP2 Annealed 1050 °C for 0.5 h**	493 ± 3	1166 ± 15	11 ± 1	10 ± 1	580	0.50
**SP2 Annealed 1100 °C for 0.5 h**	455 ± 5	1147 ± 14	17 ± 2	15 ± 1	590	0.51
**SP2 Annealed 1150 °C for 0.5 h**	436 ± 3	1083 ± 24	23 ± 1	21 ± 1	600	0.56
**Once-sintered 90** **°**						
**MP1 Annealed 1000 °C for 0.5 h**	707 ± 3	1196 ± 48	11 ± 3	11 ± 2	650	0.54
**MP1 Annealed 1150 °C for 0.5 h**	649 ± 2	1228 ± 7	36 ± 1	28 ± 1	650	0.53
**Once-sintered 45** **°**						
**SP2 Annealed 1000 °C for 0.5 h**	644 ± 7	1310 ± 7	7 ± 1	7 ± 1	690	0.53
**SP2 Annealed 1150 °C for 0.5 h**	460 ± 15	1096 ± 9	27 ± 1	22 ± 1	600	0.55
**Once-sintered 0** **°**						
**SP2 Annealed 1000 °C for 0.5 h**	635 ± 6	1344 ± 12	8 ± 1	8 ± 1	660	0.49
**SP2 Annealed 1150 °C for 0.5 h**	459 ± 12	1116 ± 8	26 ± 1	21 ± 1	550	0.49
**Hot–forged Co–Cr–Mo**						
**Annealed 750 °C for 1 h**	934 ± 12	1296 ± 4	22 ± 1	19 ± 1	1010	0.78
**Annealed 900 °C for 1 h**	909 ± 13	1360 ± 21	15 ± 1	13 ± 1	995	0.73
**Annealed 950 °C for 1 h**	850 ± 23	1308 ± 35	17 ± 1	14 ± 1	900	0.69
**Annealed 1000 °C for 1 h**	759 ± 4	1231 ± 1	22 ± 1	19 ± 1	820	0.65
**Annealed 1050 °C for 1 h**	676 ± 4	1196 ± 9	30 ± 1	25 ± 1	755	0.63
**Annealed 1100 °C for 1 h**	587 ± 1	1040 ± 10	27 ± 2	23 ± 1	655	0.63
**Annealed 1150 °C for 1 h**	538 ± 5	975 ± 6	25 ± 1	23 ± 1	555	0.57

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
