# Peer review of "Chemical, Physical, and Mechanical Properties and Microstructures of Laser-Sintered Co–25Cr–5Mo–5W (SP2) and W–Free Co–28Cr–6Mo Alloys for Dental Applications"

_materials, 2019, doi:10.3390/ma12244039_

Round 1
Reviewer 1 Report
Data are quite helpful for dental applications.
Findings are as expected and can be trusted.
Author Response
Thank you for the peer review of our manuscript. We were pleased with the comments you made on the manuscript.
Reviewer 2 Report
The article is well written. It deals with a very contemporary topic. I actually have only on comment regrading the presentation of results:
The authors have performed a lot of fatigue tests, but the corresponding data processing is not well described in section 2.6. In the article there is no clear information on how did they model the S-N curves and estimate the fatigue durability limits. Since the statistical scatter is significant (see figures 8, 9, 12 and 13), the fatigue limit should be linked with some probability of rupture. This means that it would be advantageous, if P-S-N curves were modelled, e.g. see "Klemenc J., Fajdiga M. Estimating S-N curves and their scatter using a differential ant-stigmergy algorithm. Int. J. Fatigue 2012; 43: 90–97" as the reference on that topic.
Otherwise the article is well writen.
Author Response
Thank you for the peer review of the manuscript, which has been revised in accordance with your comments. The modifications are as follows:
(1) We have added the estimation method of the fatigue durability limit to section 2.6. Specifically, the fatigue durability limit was estimated by visually estimating the maximum stress at which a specimen does not break after 107 cycles from the S-N curve as shown in Fig. 8. We tried to statistically analyze the fatigue limit using analytical software (Standard evaluation method of fatigue reliability for metallic materials-Standard regression method of S-N curves-) recommended by the Japan Society of Materials Science, but the fatigue durability limit was not able to be analyzed well. We are familiar with the P-S-N curve recommended by the reviewer, but we were unable to perform on analysis using the curve in this study. Therefore, we would like to make it a future study subject.
Reviewer 3 Report
This work deals with the microstructure and mechanical properties of alloys used in dental applications. The materials are processed by laser sintering and their properties are compared with those of the alloys produced by casting or hot forging. The authors have done a large amount of experimental work and used different methods to characterize the alloy samples. The paper is of very good quality and requires minor revision before publication.
Please revise this phrase in the Abstract: "The changes in the chemical composition after laser sintering 20 times were negligible...", otherwise it is unclear. Please consider changing the phrase "20 times sintered". Was it sintering by 20 passes (20 indentical cycles)? This should be clarified in the Abstract as well as in the text. You mention "The effect of the number of repeated uses of the same powder on these properties was also investigated." Is this intention related to the "sintering times"? How did you select the number of the "sintering times"?
The Conclusions should be revised to become more concise. In the present version, the results are mainly repeated. The comparison of the cast, hot forged and laser sintered samples should be given in a clear manner for the reader to grasp the information easily.
Could you please specify the role of tungsten in these alloys?
Author Response
Thank you for the peer review of the manuscript, which has been revised in accordance with your comments. The modifications are as follows:
(1) The phrase "The changes in the chemical composition after laser sintering 20 times were negligible..." in Abstract has been modified. We have also added the reason why we chose 20 times as the number of times of laser sintering with the same powder without adding virgin powder.
(2) We have revised the conclusion. We have also made it easier to understand the comparison of the characteristics of laser-sintered and hot-forged materials. The fourth paragraph of the conclusion was edited to make it easier for readers to understand.